# Martian World Model: Controllable Video Synthesis with Physically Accurate 3D Reconstructions

**Longfei Li**[1], **Zhiwen Fan**[2,*] **Wenyan Cong**[2], **Xinhang Liu**[3],
**Yuyang Yin**[1], **Matt Foutter**[4], **Panwang Pan**[5], **Chenyu You**[6], **Yue Wang**[7,8],
**Zhangyang Wang**[2], **Yao Zhao**[1], **Marco Pavone**[4,8], **Yunchao Wei**[1,†]

[1]BJTU  [2]UT Austin  [3]HKUST  [4]Stanford University  [5]XMU  [6]SBU  [7]USC  [8]NVIDIA

**Project Website**: https://marsgenai.github.io

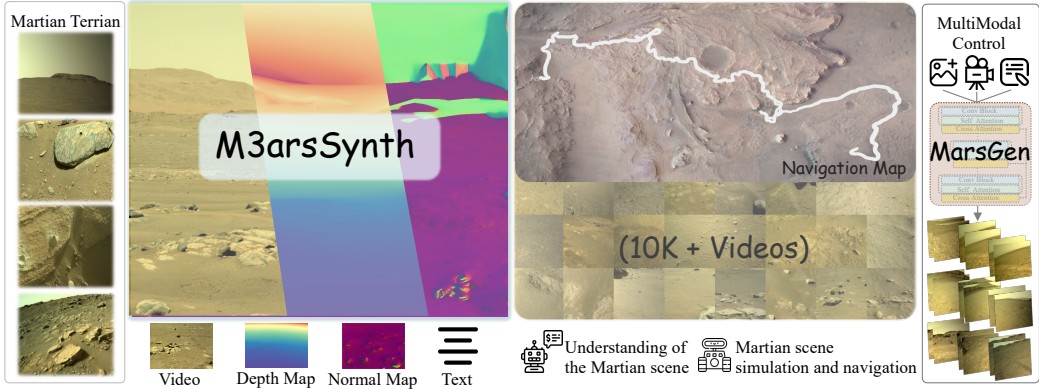

Figure 1: **Overview of the M3arsSynth data engine and MarsGen video generator.** The **M3arsSynth** engine processes NASA stereo navigation imagery into a versatile multimodal Mars dataset comprising video, depth/normal maps (from 3D reconstructions), and text descriptions. These outputs advance Mars scene generation and simulation for mission rehearsal and robotic navigation.

## Abstract

Synthesizing realistic Martian landscape videos is crucial for mission rehearsal and robotic simulation. However, this task poses unique challenges due to the scarcity of high-quality Martian data and the significant domain gap between Martian and terrestrial imagery. To address these challenges, we propose a holistic solution composed of two key components: 1) A data curation pipeline *Multimodal Mars Synthesis (M3arsSynth)*, which reconstructs 3D Martian environments from real stereo navigation images, sourced from NASA's Planetary Data System (PDS), and renders high-fidelity multiview 3D video sequences. 2) A Martian terrain video generator, *MarsGen*, which synthesizes novel videos visually realistic and geometrically consistent with the 3D structure encoded in the data. Our M3arsSynth engine spans a wide range of Martian terrains and acquisition dates, enabling the generation of physically accurate 3D surface models at metric-scale resolution. MarsGen, fine-tuned on M3arsSynth data, synthesizes videos conditioned on an initial image frame and, optionally, camera trajectories or textual prompts, allowing for video generation in novel environments. Experimental results show that our approach outperforms video synthesis models trained on terrestrial datasets, achieving superior visual fidelity and 3D structural consistency.

---

*Z. Fan is the Project Lead

†Y. Wei is the Corresponding Author

39th Conference on Neural Information Processing Systems (NeurIPS 2025) Track on Datasets and Benchmarks.

# 1  Introduction

The advancement of space exploration is critically dependent on the development of robust robotic systems and operational procedures Gao and Chien (2017) tailored to diverse extraterrestrial environments. A major challenge across such domains is the lack of platforms capable of synthesizing realistic and dynamic data. This limitation hinders autonomous mission planning Maurette (2003), operational rehearsal Wright et al. (2005), rover navigation, and the execution of complex robotic tasks Huntsberger et al. (2000); Mathers et al. (2012). Current publicly available extraterrestrial imagery, for example from NASA's Martian rovers such as Curiosity and Perseverance—is typically provided as static stereo pairs captured from discrete and sparsely distributed viewpoints. The quality of these images is also affected by the interplanetary bandwidth limitations Goldstein (1968) and operational constraints. As a result, reconstruction of photorealistic 3D environments from such sparse and constrained imagery, a common challenge in planetary datasets, remains a obstacle.

Recent advances in video synthesis Brooks et al. (2024); Kong et al. (2024); Jin et al. (2024) offer promising avenues to mitigate these challenges. However, training an effective video generation model conditioned on sparse Martian imagery or reconstructed 3D models remains difficult. Most existing models Yang et al. (2024); Wang et al. (2025a), which are typically trained on large-scale terrestrial datasets, struggle to generalize to the Martian domain due to the substantial domain gap. In addition, the challenges inherent to 3D reconstruction on Martian data often result in geometric models with insufficient accuracy, making it difficult to support high-fidelity, 3D-consistent video synthesis from such limited inputs.

To bridge this gap, we introduce a holistic solution for the synthesis of realistic Martian landscape 3D videos. We first propose **Multimodal Mars Synthesis (M3arsSynth)**, a data curation framework. M3arsSynth processes sparse and photometrically inconsistent stereo navigation images, sourced from NASA's Planetary Data System (PDS) [3]. Leveraging the strong generalization capabilities of geometric foundation models, it achieves robust 3D scene reconstruction as a critical intermediate step. This process allows for the creation of physics-accurate 3D surface models at metric-scale resolution, which form the basis for rendering high-fidelity 3D video sequences. The principal output of M3arsSynth is a large-scale, versatile multimodal dataset. This dataset includes synthesized videos, corresponding camera motion trajectories, detailed geometric information such as depth maps, and associated textual descriptions, all designed for diverse Martian applications. The second component of our solution is **MarsGen**, a video-based Martian terrain generator. MarsGen utilizes the rich dataset produced by M3arsSynth along with other multimodal conditioning inputs (such as an initial image frame, specified camera trajectories, or textual prompts) to accurately synthesize novel, 3D-consistent video frames and dynamic environments, enabling the controllable generation of new Martian scenarios not present in the original rover data.

Experimental results demonstrate that our unified solution significantly surpasses existing Earth-trained video synthesis approaches, encompassing both open-source and closed-source alternatives. Our method achieves superior visual quality in the generated Martian videos, robust 3D consistency across frames, and enhanced camera controllability, offering a substantial improvement for realistic simulation. Our primary contributions are:

- We introduce M3arsSynth, a multimodal data engine that transforms challenging rover-captured stereo navigation imagery into high-quality assets for synthesizing controllable video for Mars missions. By leveraging geometric foundation models, M3arsSynth creates metric-scale 3D environments, effectively addressing critical issues such as sparse-view coverage and photometric inconsistencies to produce over 10K physically accurate 3D Martian surface models.

- Our work enables controllable video generation of Martian terrain, MarsGen, starting from a single-view image input and conditioned on camera poses or text prompts, yielding photorealistic and 3D-consistent video sequences.

- We evaluate our generated controllable video across key metrics, including visual fidelity, our proposed 3D video consistency, and camera controllability. Our approach significantly outperforms models trained primarily on terrestrial data, demonstrating its strong potential for future data-driven robotic simulation.

---

[3]https://pds-imaging.jpl.nasa.gov/beta/archive-explorer

## 2 Related works

**Planetary Environment Simulation.** Prior efforts in simulating planetary environments Jain et al. (2003); Tian et al. (2024) have focused on enhancing the interpretation and interaction with Martian data. Approaches include Mixed Reality (MR) technologies Mahmood et al. (2019); Memarsadeghi and Varshney (2020), such as those developed by NASA JPL's Operations Lab and Microsoft Abercrombie et al. (2017); Beaton et al. (2020), enabling immersive exploration of 3D terrain models from rover data. Additionally, 3D reconstruction techniques like the MaRF Giusti et al. (2022) framework, which employs Neural Radiance Fields (NeRF) Mildenhall et al. (2021) for continuous volumetric representations from sparse images, have improved visualization from novel viewpoints. However, the MaRF framework is notably limited, having been demonstrated in different 3D environments. More broadly, these existing technologies often require high-quality, consistent visual data and exhibit limitations in scalability and adaptability. Our approach addresses these challenges by employing video generation models trained on a dedicated video dataset produced by our specialized data processing pipeline.

**3D Modeling from Sparse Views.** Reconstructing 3D scenes from sparse views Chen and Wang (2024) is a significant challenge. NeRF and 3DGS Kerbl et al. (2023) typically demand hundreds of images and rely on the Structure-from-Motion (SfM) Schönberger and Frahm (2016) approach (e.g., COLMAP Schonberger and Frahm (2016)). To address this, some works leverage priors by pre-training on large datasets Chen et al. (2021); Johari et al. (2022); Yu et al. (2021); Chibane et al. (2021); Jang and Agapito (2021) or by applying regularization during NeRF optimization Wang et al. (2023); Roessle et al. (2023); Seo et al. (2023); Somraj et al. (2023); Somraj and Soundararajan (2023); Wynn and Turmukhambetov (2023); Liu et al. (2024). To mitigate the overfitting to input views in 3DGS, FSGS Zhu et al. (2024) and SparseGS Xiong (2024) incorporate external priors from depth estimator with the optimization process. Others, like InstantSplat Fan et al. (2024), utilize powerful 3D reconstruction models Leroy et al. (2024) to acquire accurate camera poses and initial geometries. However, robust sparse-view reconstruction remains an open problem, particularly in challenging environments such as the Martian surface, which exhibit textureless areas, repetitive patterns, and photometric variations. Our method addresses this gap by integrating a 3D geometric foundation model for initial geometric estimation with a specialized pipeline for refinement and neural scene representation from sparse stereo imagery, facilitating high-quality multimodal data synthesis.

**Conditional Generative Models.** Recent text-to-video models Yang et al. (2024); Kong et al. (2024); Wang et al. (2025a); Blattmann et al. (2023) based on Diffusion Transformers (DiTs) Peebles and Xie (2023) leverage the scalability of transformers, typically employing text encoders Radford et al. (2021); Raffel et al. (2020), a 3D-VAE Yu et al. (2023); Yang et al. (2024) for video compression and tokenization, and a transformer generator that processes flattened video and text tokens. These architectures model spatiotemporal and textual information through global attention mechanisms or separate self- and cross-attention, leading to significant improvements in generation duration and temporal consistency. View-controllable video generation Wang et al. (2024c); He et al. (2024); Liang et al. (2024); Bahmani et al. (2024); Yu et al. (2024), which is crucial for immersive simulations, has seen efforts to integrate camera control into pretrained models. However, these methods, predominantly trained on standard terrestrial datasets, often exhibit difficulties with 3D consistency when applied to out-of-domain environments like Mars. In contrast, our work utilizes the M3arsSynth dataset, specifically curated with rich 3D geometric information, to train MarsGen, enabling physically plausible and precisely controllable Martian simulations with enhanced 3D consistency.

## 3 Multimodal Mars Synthesis

We establish the M3arsSynth dataset from curated rover stereo image from NASA PDS. To render a large-scale multimodal dataset suitable for training generative simulators, we leverage robust, pre-trained vision foundation models to capture Martian visual cues and compensate for the lack of Mars-specific priors. In this section, Sec. 3.1 first describes the source data acquisition and preprocessing steps. Sec. 3.2 then outlines the metric-aware 3D reconstruction process. Sec. 3.3 details the synthesis of multimodal data, and summarizes the overall structure of the dataset. Finally, Sec. 3.4 explains the proposed 3D consistency metrics for evaluating generated video sequences.

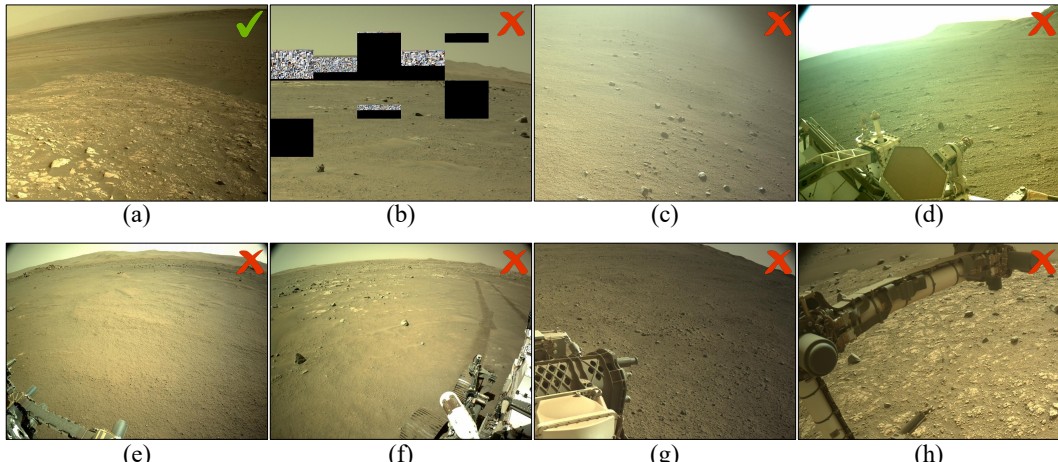

Figure 2: **Data Filtering Examples for Martian 3D Reconstruction.** Image (a) represents a clear, usable Martian terrain view, serving as a quality benchmark. In contrast, images (b)-(h) illustrate common defects that lead to data exclusion for high-quality reconstruction, including: (b) extensive missing data blocks or pixelation/mosaic artifacts (indicative of data corruption or severe compression); (c) significant image blur or out-of-focus areas; (d) scenes with extreme overexposure or harsh lighting conditions; and (e)-(h) views obstructed by spacecraft components.

## 3.1 Martian Stereo Image Acquisition and Preprocessing

Raw stereo image pairs captured by Martian rovers are sourced from PDS. This initial dataset, however, exhibits several deficiencies, including small thumbnail images, grayscale images, and duplicate captures made with different color filters, rendering it unsuitable for direct application. To address these issues, an automated filtering pipeline is implemented. This pipeline is designed to systematically remove these types of deficient data:

**Systematic Data Filtering Strategy.** A multi-stage filtering pipeline ensures visual dataset integrity by systematically addressing imperfections. Firstly, *low-resolution and grayscale images are eliminated*. Thumbnails are discarded based on size heuristics. Grayscale images are removed based on low RGB channel variance. Secondly, *redundant content is excluded using perceptual hashing* Zauner (2010). This technique employs hash codes and Hamming distances to identify and remove near-duplicates captured under varied imaging conditions, preserving semantic diversity. Thirdly, *blurry and low-sharpness images are rejected*. A sharpness filter using Laplacian variance Bansal et al. (2016) discards images with low edge contrast to maintain geometric and photometric quality, which is vital for tasks like stereo reconstruction. Finally, *frames exhibiting anomalous color distributions are filtered out*. Irrelevant frames (e.g., obscured, malfunctions), identified by skewed or flat color intensity histograms, are removed to ensure clear, terrain-focused scenes for robust surface representation learning.(See Appendix A.1 for details.)

**Semi-Automated Refinement.** To address complex visual artifacts like hardware occlusions and unfavorable lighting that degrade reconstruction quality (see Fig. 2), we employ a semi-automated refinement process. This step supplements manual image verification with Grounded-SAM Ren et al. (2024), which we use to generate segmentation masks identifying non-terrain objects based on textual prompts. These masks guide human annotators to efficiently identify and discard compromised image data. The result is a curated collection of images with high visual integrity, providing a reliable foundation for robust stereo reconstruction. Further details on this preprocessing are provided in Appendix A.2.

## 3.2 Neural 3D Reconstruction from Navigation Stereo Cameras

Dense Martian 3D reconstruction poses unique challenges compared to terrestrial scenes, stemming from the planet's often texture-poor terrain and the inherent scarcity of observational data.

**Camera Calibration.** Our 3D reconstruction pipeline operates on the standard pinhole camera model Hartley (2003) and therefore requires the corresponding camera parameters. However, the PDS metadata accompanying our dataset lacks these crucial calibration parameters. To address this limitation, we infer the necessary parameters directly from the images. We employ the Visual Geometry Grounded Transformer (VGGT) Wang et al. (2025b), a feed-forward neural network

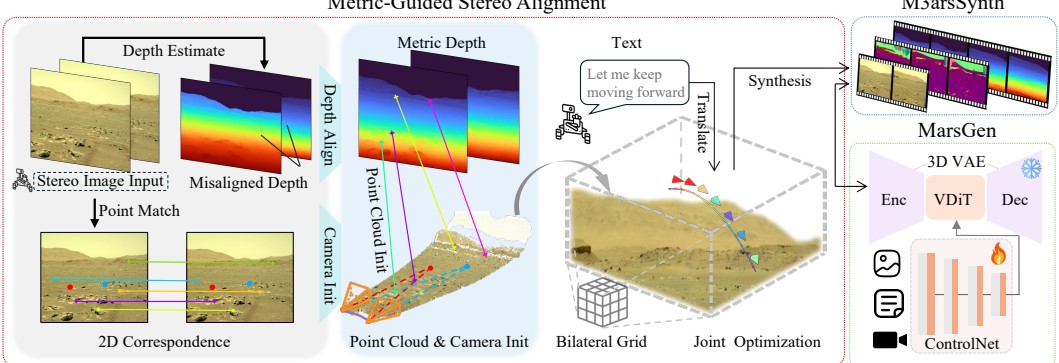

Figure 3: **Overview of the M3arsSynth dataset construction and conditional video generation through MarsGen**. The red box outlines the data curation pipeline, the green box shows the obtained M3arsSynth dataset, and the blue box details our MarsGen model. We process stereo image pairs using a metric-aware foundation model and solve the Perspective-n-Point (PnP) Lepetit et al. (2009) problem to reconstruct metric-scale 3D Martian scenes. Subsequently, video frames rendered from these scenes, together with text prompts and encoded camera trajectories, are then used to condition a Video Diffusion Transformer, enabling the synthesis of novel and controllable Martian video sequences.

designed to infer key 3D scene attributes from input views. Our empirical evaluations confirm that VGGT reliably predicts the camera intrinsics from the input stereo images, providing the essential parameters for subsequent reconstruction tasks.

**Dense 3D Geometry Initialization.** After obtaining camera intrinsics, we estimate metric deoth maps using a pretrained monocular depth estimation network Hu et al. (2024) to construct dense 3D geometry. Each depth map is then back-projected using the inverse intrinsic matrix $K^{-1}$ to transform each pixel $(u, v)$ with depth $d$ into a 3D point in the corresponding camera coordinate system: $P_c = d \cdot K^{-1}[u, v, 1]^T$. This process yields initial per-view point clouds where each pixel has a direct 3D correspondence within its respective camera frame, providing a dense geometry initialization.

**Relative Pose Estimation and Geometric Refinement.** To establish a coherent 3D model, we recover the relative camera extrinsics between the stereo images. We formulate this task as a Perspective-n-Point (PnP) Li et al. (2012) optimization, as PnP computes the camera pose from a set of 3D points and their corresponding 2D image projections. This approach utilizes robust 2D feature correspondences $\mathbf{p}_1 \leftrightarrow \mathbf{p}_2$ between the stereo image pair, which are detected using the Generalizable Image Matcher (GIM) Shen et al. (2024). GIM's notable generalization capability stems from its training on extensive and varied internet video data (approximately 180,000 image pairs from 50 hours of video) and its sophisticated self-training framework. This enables robust matching across diverse conditions and often surpasses traditional methods that rely on less diverse datasets or failure-prone 3D reconstruction processes Shen et al. (2024). Here, $\mathbf{p}_1$ and $\mathbf{p}_2$ represent matched pixel coordinate vectors in the left and right views, respectively. Initial depth estimates from the monocular network are used to back-project points $\mathbf{p}_1$ into 3D space, yielding a set of 3D points $\mathbf{P}_1$. Given the known camera intrinsics $K$, these 2D-3D correspondences (formed by $\mathbf{P}_{1,i}$ and their corresponding $\mathbf{p}_{2,i}$) facilitate the estimation of the relative camera pose.

The PnP solver estimates the rotation $R_{\text{rel}}$ and translation $\mathbf{t}_{\text{rel}}$ that best align the 3D points $\mathbf{P}_{1,i}$ with their corresponding 2D projections $\mathbf{p}_{2,i}$ in the second view by minimizing the reprojection error:

$$\min_{R_{\text{rel}}, \mathbf{t}_{\text{rel}}} \sum_i \|\mathbf{p}_{2,i} - \pi\left(K[R_{\text{rel}} \mid \mathbf{t}_{\text{rel}}]\mathbf{P}_{1,i}\right)\|^2, \tag{1}$$

where $\pi$ denotes the perspective projection from the 3D camera coordinates to the 2D pixel space. The initial geometry derived from the monocular depth estimation may exhibit inconsistencies, particularly in scale across views. To ensure a coherent 3D reconstruction, we apply a depth rescaling step after pose estimation. Specifically, we first back-project the depth map $D_0$ from view 0 to reconstruct its corresponding 3D point cloud $P_0$. Then we project it onto view 1's image plane. This warping process yields a new sparse depth map $D'_{0 \to 1}$ and a Boolean mask $M_{sparse}$, indicating valid

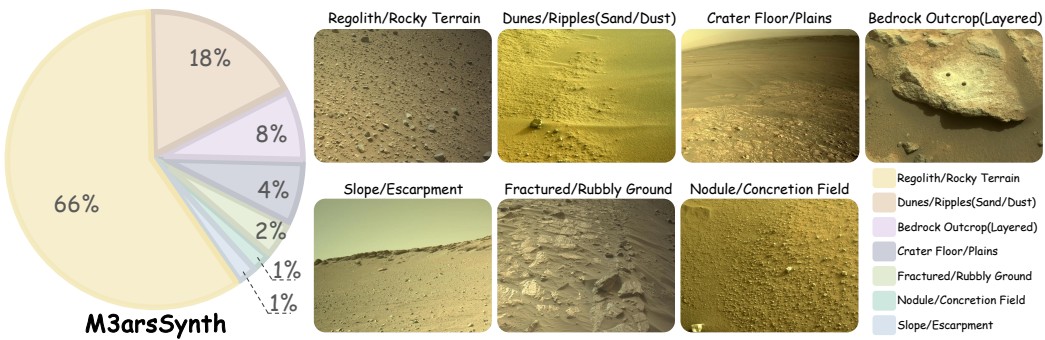

Figure 4: Distribution of primary terrain types within the M3arsSynth dataset, showcasing the diversity of Martian environments covered. The left chart indicates the percentage of scenes predominantly featuring each terrain type, with visual examples illustrating the various terrain categories.

projection areas in view 1. The depth value $d'_1$ is stored in $D'_{0 \to 1}$ at the projected pixel coordinates $(u'_1/d'_1, v'_1/d'_1)$.

$$P_0 = D_0(p_0) \cdot K_0^{-1} \begin{bmatrix} u_0 \\ v_0 \\ 1 \end{bmatrix}$$

$$\begin{bmatrix} u'_1 \\ v'_1 \\ d'_1 \end{bmatrix} = K_1(RP_0 + t)$$

Then, we align the original monocular depth map of view 1 $D_1$, with the warped sparse depth map $D'_{0 \to 1}$. We solve a least-squares regression problem only within the valid region defined by the mask $M_{sparse}$.

$$\min_{s,b} \sum_{(u,v) \in M_{sparse}} (s \cdot D_1(u,v) + b - D'_{0 \to 1}(u,v))^2$$

This estimated $(s, b)$ are then used to adjust the depth map of the first view (i.e., $d_{1,j}^{\text{adjusted}} = s \cdot d_{1,j} + b$) to better match the scale and offset of the second view's depth map, or vice-versa, thereby enhancing geometric consistency.

**3D Gaussian Splatting for Photorealistic Scene Modeling.** Using the camera parameters and dense point cloud from prior stages, we optimize a 3DGS Kerbl et al. (2023); Huang et al. (2024) representation, initializing Gaussian primitives directly from the per-pixel points Fan et al. (2024). The optimization combines a photometric loss with a depth regularization loss Xu et al. (2025); Xiong (2024); Li et al. (2024); Kerbl et al. (2024) that leverages our derived per-pixel correspondences to enforce geometric consistency. To mitigate significant appearance variations between Martian stereo pairs, often caused by differing camera settings or lighting, we integrate a bilateral grid Wang et al. (2024b) into the process. Specifically, our implementation is based on gsplat Ye et al. (2025). We apply a per-view 3D bilateral grid as a differentiable post-processing layer to the rendered image, which models view-dependent effects. This grid is jointly optimized with the Gaussian parameters by minimizing the difference between the post-processed render and the corresponding training view, while a total variation loss is used to regularize the grid for smoothness. Furthermore, since standard k-nearest neighbor initialization can produce overly large Gaussian scales and degrade depth accuracy, we adopt a modified strategy Cong et al. (2025) where scale is determined by the nearest point along the depth axis.

$$\text{scale}(P_w) = d'_{\min}(P_w)/f_{\text{avg}}$$

### 3.3 Multimodal Martian Dataset Creation

Upon obtaining an optimized neural scene representation for a Martian scene, we sample a diverse set of virtual camera trajectories, denoted $\mathcal{M}_{\text{traj}}$, along which video sequences are rendered:

$$\mathcal{M}_{\text{traj}} = \left\{ T_t = \begin{pmatrix} \mathbf{R}_t & \mathbf{T}_t \\ \mathbf{0}_{1 \times 3} & 1 \end{pmatrix} \in SE(3) \mid t = 1, \dots, N \right\} \tag{2}$$

where each transformation $T_t$ defines the 6-DOF camera pose at timestamp $t$, comprising a rotation $\mathbf{R}_t$ and a translation $\mathbf{T}_t$. Canonical trajectory types are defined, encompassing diverse motion profiles. Their spatial extent is adapted to the scene geometry through depth-adaptive scaling: trajectories are

contracted for near-field regions to capture fine details and expanded for far-field regions to facilitate wider movements. Further details regarding the specific parameters and generation process for these trajectories are provided in the Supplementary Material.

**Video, Normal, Trajectory, Depth, Text.** From a set of adaptively scaled camera trajectories, we generate a comprehensive multimodal dataset. Novel view videos, accompanied by their corresponding depth and normal maps, are produced by rendering the scene along these trajectories. The Trajectory modality encompasses the precise 6-DOF pose parameters for each frame ($T_t$), from which textual descriptions of camera motion are subsequently derived. To capture scene content, we further generate textual captions by applying a Vision Language Model (VLM) OpenAI (2024) to the rendered videos. This procedure results in a rich dataset comprising visual, geometric, and textual modalities, all structured around the foundational camera trajectories.

**Dataset Structure.** The resulting M3arsSynth dataset consists of a diverse set of distinct Martian scenes, each reconstructed and rendered from an optimized neural scene representation. For each scene, we generate multiple video sequences, each simulating a unique virtual camera trajectory. These sequences are temporally structured and provide rich, time-aligned multimodal information. Specifically, each sequence contains: (i) synthesized RGB frames rendered from novel viewpoints, (ii) precise 6-DOF camera poses corresponding to each frame, (iii) natural language descriptions detailing both the visual content and camera motion characteristics, and (iv) corresponding per-frame geometric outputs, including depth maps and optionally surface normal maps.

## 3.4 Metrics Evaluating 3D Consistency

To assess the 3D consistency of video sequences generated by our MarsGen model (trained on the M3arsSynth dataset), we employ the *2D Warp Error* metric Wang et al. (2024a).

The 2D Warp Error measures the internal geometric consistency of the 3D structure implicitly represented in the generated video frames. Specifically, this metric evaluates how accurately 3D points, inferred from each generated frame, are reprojected onto other frames or consistently within the same frame. The closer these reprojected points align with their corresponding expected 2D locations (e.g., on a canonical grid), the more consistent the underlying geometry is considered. This evaluation comprises two main components:

**Self-Reprojection Consistency.** For each generated frame $i$ within a sequence, an associated 3D point cloud $\mathcal{P}_i^{(c)}$ (composed of points $\mathbf{p}_j^{(c)}$) in its local camera coordinate system, along with the camera intrinsic matrix $K_i$, is typically inferred using a geometric perception model applied to that frame. Each 3D point $\mathbf{p}_j^{(c)} \in \mathcal{P}_i^{(c)}$ is then projected onto the 2D image plane of frame $i$ using $K_i$, resulting in reprojected pixel coordinates $\mathbf{x}_{\text{reproj},j}$. The self-reprojection consistency for frame $i$ is computed as the mean squared Euclidean distance between these reprojected coordinates and their expected positions on a canonical 2D sampling grid:

$$L_{\text{self-reproj},i} = \frac{1}{M} \sum_j \|\mathbf{x}_{\text{reproj},j} - \mathbf{x}_{\text{gtgrid},j}\|_2^2 , \tag{3}$$

where $M$ denotes the number of points in $\mathcal{P}_i^{(c)}$, and $\mathbf{x}_{\text{gtgrid},j}$ is the source grid location of point $j$.

**Cross-View Reprojection Consistency.** To evaluate geometric consistency across different viewpoints, we measure the cross-frame reprojection between frame $i$ and another frame $k$ from the same sequence. First, we estimate the 3D point cloud $\mathcal{P}_k^{(c)}$ (composed of points $\mathbf{p}_{j'}^{(c)}$) for frame $k$ in its camera coordinate system, along with the relative transformation $M_{i \leftarrow k} \in SE(3)$ mapping points from frame $k$'s coordinate system to that of frame $i$. Each point $\mathbf{p}_{j'}^{(c)} \in \mathcal{P}_k^{(c)}$ is transformed into the coordinate space of frame $i$:

$$\mathbf{p}_{j'}^{(c,i \leftarrow k)} = M_{i \leftarrow k} \mathbf{p}_{j'}^{(c)} .$$

These transformed 3D points are then projected onto the image plane of frame $i$ using its intrinsic matrix $K_i$, yielding reprojected pixel coordinates $\mathbf{x}_{\text{reproj},j'}$. The cross-view reprojection loss between frames $i$ and $k$ is computed as:

$$L_{\text{cross-reproj},i,k} = \frac{1}{M'} \sum_{j'} \|\mathbf{x}_{\text{reproj},j'} - \mathbf{x}_{\text{gtgrid},j'}\|_2^2 , \tag{4}$$

Table 1: **Quantitative comparison of video generation models.** We evaluate visual fidelity (FID, FVD), 3D consistency (Warp Error), and novel view synthesis quality (PSNR, SSIM, LPIPS). For these metrics, lower values indicate better performance for FID, FVD, Warp Error, and LPIPS, while higher values are preferable for PSNR and SSIM. Our MarsGen demonstrates state-of-the-art results across all evaluated metrics.

| Model | Visual Fidelity | | 3D Consistency | Novel View Synthesis | | |
|---|---|---|---|---|---|---|
| | FID ↓ | FVD ↓ | Warp Err↓ | PSNR ↑ | SSIM ↑ | LPIPS ↓ |
| *Image-to-Video* | | | | | | |
| Pyramidal-Flow Jin et al. (2024) | 78.495 | 637.952 | 17.930 | – | – | – |
| CogVideoX Yang et al. (2024) | 48.912 | 411.808 | 6.866 | – | – | – |
| Kling | 74.632 | 727.130 | 24.793 | – | – | – |
| Sora Brooks et al. (2024) | 142.954 | 823.418 | 10707.713 | – | – | – |
| *Camera Control Image-to-Video* | | | | | | |
| CameraCtrl He et al. | 123.386 | 772.476 | 17.410 | 20.014 | 0.441 | 0.408 |
| ViewCrafter Yu et al. (2024) | 169.942 | 2297.899 | 501.734 | 13.143 | 0.500 | 0.586 |
| Ours | **38.779** | **364.822** | **6.071** | **21.239** | **0.614** | **0.351** |

where $M'$ is the number of points in $\mathcal{P}_k^{(c)}$, and $\mathbf{x}_{\text{gtgrid},j'}$ denotes the expected projection location in frame $i$ (e.g., corresponding to a canonical grid position from frame $k$). This metric captures how well the geometry inferred from one frame generalizes across viewpoints, revealing discrepancies in spatial alignment and structure.

The overall 2D Warp Error reported is typically an average of these component losses (e.g., $L_{\text{self-avg}}$ and $L_{\text{cross-avg}}$ being the mean self-reprojection and cross-view reprojection errors, respectively) over the sequence or relevant frame pairs:

$$L_{\text{2D-Reproj}} = \frac{1}{2} \left( L_{\text{self-avg}} + L_{\text{cross-avg}} \right).$$

A lower reprojection error indicates better geometric consistency.

## 4 Martian Terrain Video Generator Training

With multimodal M3arsSynth dataset curated by our data engine, we develop and train MarsGen, a conditional generative model for Martian video synthesis. The primary goal of MarsGen is to produce novel, photorealistic video sequences of Martian environments that are not only visually compelling but also exhibit high levels of 3D geometric consistency and physical plausibility. Conditioned on textual prompts or predefined camera trajectories, MarsGen generates temporally coherent RGB sequences that align with the underlying scene structure inferred from the conditioning inputs.

**Model Architecture.** MarsGen is built upon the Video Diffusion Transformer (VDiT) framework Peebles and Xie (2023). The video and text prompts are first encoded into a latent space Yu et al. (2023); Raffel et al. (2020) and concatenated. A 3D self-attention mechanism, operating across both temporal and spatial dimensions, facilitates a comprehensive interaction between the multimodal information. Finally, a decoder reconstructs the latent representations back into the video space. We train this model on the M3arsSynth dataset to adapt it to the unique visual cues and structural characteristics of Martian environments, thus supporting high-fidelity and controllable video generation under domain-specific constraints.

**Controllable Content Generation.** To achieve fine-grained control over the generative process, MarsGen incorporates multiple modalities, including a reference initial frame, textual prompts (natural language descriptions), and camera trajectories. These inputs jointly guide the model across spatial, temporal and semantic dimensions. To circumvent the computational demands of full model fine-tuning, we employ a lightweight conditioning strategy inspired by ControlNet architecture Zhang et al. (2023). Specifically, we inject control signals into intermediate layers of the pretrained VDiT backbone, enabling the modulation of generation dynamics without disrupting its learned visual priors.

**Video Model Training.** We initialize the VDiT backbone with pretrained weights from CogVideoX-5B-I2V Yu et al. (2023); Yang et al. (2024). We then fine-tune the model, incorporating its controllable branch, on 8 A100 GPUs for 8,000 iterations using the M3arsSynth dataset.

## 5 Experiments

We evaluate the MarsGen generator, trained on our MarsSynthSim dataset, through comprehensive quantitative and qualitative experiments. This section further compares different reconstruction

Table 2: **Quantitative comparison of reconstruction pipelines for Martian stereo imagery.** We compare COLMAP, the Transformer-based MASt3R Leroy et al. (2024), and our metric-aware initialization method across four key metrics: runtime efficiency (Time), frame-level robustness (Data Utilization), geometric accuracy (2D Reprojection Error Wang et al. (2024a)), and reconstruction density (Point Number). Our pipeline fully utilizes all available data and achieves competitive accuracy, whereas COLMAP fails on nearly 30% of the preprocessed image pairs.

| Algorithm | Data Util. (%) | Time (s) | Reproj Err (px) ↓ | Point Num |
|---|---|---|---|---|
| COLMAP Schönberger and Frahm (2016); Schönberger et al. (2016) | 71.8 | **3.6** | **0.134** | $\sim 2,000$ |
| MASt3R Leroy et al. (2024) | 100.0 | 8.7 | 46.98 | $\sim 250,000$ |
| Ours | **100.0** | 8.0 | 0.77 | $\sim 250,000$ |

methods used in the creation of the MarsSynthSim dataset and presents an ablation study of our M3arsSynth pipeline's key components, detailing their impacts.

**Martian Terrain Video Generation.** We quantitatively evaluate the performance of MarsGen, our model fine-tuned on the MarsSynthSim dataset, with a focus on both visual fidelity and 3D geometric consistency. We assess visual quality using FID (Fréchet Inception Distance) Heusel et al. (2017) and FVD (Fréchet Video Distance) Unterthiner et al. (2018), and evaluate geometric consistency through the 2D Warp Error and PSNR. A key capability of MarsGen is its precise camera pose control, which enables us to directly evaluate its novel view synthesis (NVS) performance. Table 1 compares MarsGen against state-of-the-art image-to-video and camera-controlled video generation models. MarsGen consistently outperforms all baselines in terms of visual fidelity, achieving the lowest FID and FVD. In terms of geometric consistency, MarsGen achieves the lowest 2D Warp Error and competitive PSNR, indicating superior preservation of 3D structure during generation. For novel view synthesis, MarsGen also shows superior performance across all metrics, demonstrating the benefit of 3D-aware training on MarsSynthSim. For a qualitative comparison, please see Appendix B. In contrast, general-purpose video models often struggle to maintain consistent spatial geometry under camera motion due to the absence of explicit 3D supervision.

**Geometry Initialization Methods.** We evaluate alternative approaches for initializing scene geometry within our M3arsSynth pipeline (used to generate the MarsSynthSim dataset), comparing traditional Structure-from-Motion (SfM) pipelines (e.g., COLMAP) and recent transformer-based methods (e.g., MASt3R) against our metric-aware initialization. We use the 2D reprojection error which measures the pixel distance between an observed point in the original image and its corresponding 3D point when re-projected back onto the image using the estimated camera parameters. It thereby jointly assesses the geometric accuracy of the 3D model and the estimated camera pose. Our method combines pre-trained vision foundation models to robustly extract camera and depth priors from challenging Martian stereo imagery. As indicated in Table 2, our pipeline achieves 100% data utilization and reconstructs dense point clouds while maintaining competitive

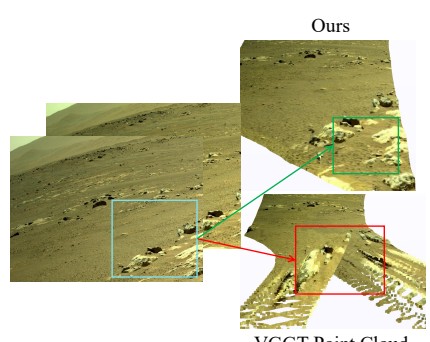

Figure 5: Qualitative comparison of point cloud reconstruction from a Martian input view. Our M3arsSynth engine (top right) produces a coherent point cloud accurately capturing terrain. In contrast, the VGGT Wang et al. (2025b) model (bottom right) exhibits significant misalignment and artifacts.

reprojection accuracy and a reasonable runtime. In contrast, COLMAP experiences partial frame failures, and MASt3R, despite generating dense reconstructions, exhibits significant reprojection errors, potentially due to overfitting to unreliable depth priors. Furthermore, we observed that point clouds derived directly from the VGGT model often contain significant artifacts (see Figure 5). We attribute these artifacts to out-of-distribution challenges encountered by the model. Consequently, our reconstruction pipeline utilizes VGGT for intrinsic estimation rather than for direct point cloud export for 3D-GS initialization.

**Ablation Studies.** To assess the contribution of individual components within our M3arsSynth reconstruction pipeline, we conduct ablation studies, with results presented in Table 3. Specifically, we evaluate two aspects: (1) the impact of Depth Rescaling, a normalization step designed to align predicted monocular depths with stereo geometry; and (2) the effectiveness of our metric-scale

Table 3: **Ablation study on the core components of the MarsSynthSim reconstruction pipeline.** This study assesses the impact of omitting depth rescaling and replacing our metric-aware initialization with a MASt3R and 3DGS baseline. The findings confirm that both depth normalization and geometric supervision are crucial for high-fidelity, structurally consistent video synthesis.

| Depth Rescaling | 3D Reconstruction | PSNR ↑ | SSIM ↑ | LPIPS ↓ |
|:---:|:---:|:---:|:---:|:---:|
| ✗ | Metric-aware Initial | 32.20 | 0.93 | 0.10 |
| ✓ | MASt3R Leroy et al. (2024) | 28.77 | 0.59 | 0.24 |
| ✓ | Metric-aware Initial | **32.73** | **0.93** | **0.09** |

reconstruction pipeline, compared against a variant that initializes geometry using MASt3R with 3DGS for rendering.

The results demonstrate that omitting depth rescaling leads to degraded rendering quality across all metrics, particularly for LPIPS, thereby highlighting the importance of geometric normalization. Furthermore, replacing our metric-scale reconstruction pipeline with the MASt3R + 3DGS baseline results in substantial reductions in both structural similarity (SSIM) and perceptual quality. This outcome underscores the value of our integrated reconstruction strategy, which is crucial for generating datasets that enable consistent and high-fidelity 3D-aware video synthesis.

# 6 Conclusion, Limitation, and Broader Impact

We introduced M3arsSynth for creating multimodal datasets from sparse rover imagery and MarsGen for generating controllable, photorealistic Martian videos. MarsGen achieves superior visual fidelity and 3D consistency over existing methods, significantly advancing Mars mission simulations for navigation and planning. While our approach significantly supports Mars mission simulation, navigation, and planning through realistic video synthesis, its current limitations include integrating predictive temporal environmental modeling and achieving fine-grained 3D semantic understanding.

**Broader Impact.** This research significantly enhances Mars mission preparedness by enabling realistic training and system testing through dynamic environmental simulations. However, the underlying generative technology also presents risks such as potential misuse for creating deceptive content, fostering over-reliance on simulations, and concerns regarding resource intensiveness and autonomous system safety.

**Acknowledgements**

This work was partially supported by Blue Origin and Redwire, as members of the Stanford Center for Aerospace Autonomy Research (CAESAR).

Yue Wang acknowledges generous supports from Toyota Research Institute, Dolby, Google DeepMind, Capital One, Nvidia, and Qualcomm. Yue Wang is also supported by a Powell Research Award.

This research has been supported by computing support on the Vista GPU Cluster through the Center for Generative AI (CGAI) and the Texas Advanced Computing Center (TACC) at the University of Texas at Austin.

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

# A   Additional Implementation Details

## A.1   Automated Data Filtering Strategies

This subsection details the automated filtering pipeline applied to the raw Martian stereo image pairs obtained from the NASA Planetary Data System (PDS). The following paragraphs describe the specific filtering techniques employed:

**Filtering Low-Quality Thumbnails and Grayscale Images.** To eliminate uninformative or non-representative visual data, we begin by removing low-resolution thumbnails and grayscale images. This step relies on file-level heuristics including image resolution, file size, and RGB channel statistics. Images with dimensions significantly smaller than our expected minimum resolution or with anomalously small file sizes are classified as thumbnails and excluded. Additionally, we assess the variance across the RGB channels to detect grayscale images; those with minimal inter-channel variance are removed, as they lack the color information necessary for downstream multimodal analysis.

**Removing Redundant Content via Perceptual Hashing.** To prevent content-level redundancy caused by multiple captures of the same scene under different imaging conditions—such as varied white balance, contrast, or filters—we apply perceptual hashing. This technique generates compact hash codes that encode high-level structural similarity. By computing Hamming distances between these hashes, we identify visually near-duplicate images and discard those exceeding a similarity threshold. This step ensures the dataset maintains semantic diversity and avoids overrepresentation of particular scenes or textures.

**Excluding Blurry and Low-Sharpness Images.** Maintaining geometric fidelity is critical for tasks such as stereo reconstruction and surface normal estimation. To this end, we apply a sharpness filter using Laplacian variance, a well-established metric that quantifies edge contrast within an image. Frames with a variance below a pre-defined threshold are classified as blurry and automatically excluded. These typically result from motion blur or poor focus, and retaining them would degrade the quality of both photometric and geometric outputs in the synthesized dataset.

**Filtering Out Visually Unusable Frames.** We analyze the average color intensity histograms of each image to identify those dominated by irrelevant content such as large spacecraft segments, occlusions, or camera malfunctions. These images often display skewed or flat histogram profiles, indicating uniform color patches or unnatural saturation patterns. We flag and remove such frames to ensure that the final dataset primarily comprises clear, terrain-focused scenes with minimal visual obstruction. This enhances the quality and consistency of the data available for learning meaningful Martian representations.

## A.2   Implementation of Grounded-SAM-assisted Semi-Automated Data Preprocessing

While the initial automated filtering pipeline, as described in Sec. A.1, addresses common image deficiencies, a subsequent semi-automated refinement stage is crucial for tackling more nuanced and complex visual challenges. These challenges include, but are not limited to, partial occlusions by rover hardware elements (e.g., wheels, antennas, or calibration targets), subtle lens-induced distortions not captured by generic filters.

This refinement phase incorporates a rigorous manual verification protocol for the image set that has passed the initial automated screening. The efficiency and accuracy of this manual review are substantially enhanced by leveraging the capabilities of Grounded-SAM, a sophisticated vision-language segmentation model. Grounded-SAM's strength lies in its ability to perform open-vocabulary segmentation, identifying and delineating image regions based on arbitrary textual prompts. This is particularly advantageous for our application, as it allows for the flexible identification of diverse and potentially unforeseen rover components or artifacts without requiring model retraining or predefined class lists.

The operational workflow is as follows:

1. **Prompt Formulation:** Human domain experts, familiar with the rover's morphology and common imaging configurations, formulate targeted textual prompts. These prompts typically reference known spacecraft components that have a high likelihood of intruding

into the image frame (e.g., "rover wheel", "robotic arm telemetry cable", "mast shadow on terrain").

2. **Mask Generation:** Grounded-SAM processes each image in conjunction with these prompts to generate segmentation masks. These masks highlight regions within the image that correspond to the textual descriptions, effectively flagging areas suspected of containing non-terrain elements or problematic features.

3. **Guided Manual Annotation:** The generated masks serve as precise visual guides for human annotators. Instead of scrutinizing the entirety of each image for potential issues, annotators can focus their attention on the regions highlighted by Grounded-SAM. This significantly accelerates the review process and improves the consistency of identifying obscured or compromised data.

Human annotators then perform the critical verification step. Based on the Grounded-SAM-proposed masks and their own expert assessment, they make the final decision to:

- Confirm and accept the mask, leading to the flagging of the highlighted region for exclusion from 3D reconstruction inputs.

- If the segmentation of Grounded-SAM is inaccurate or the image contains unrecognized errors, manual annotation is carried out to obtain a clean image.

- Flag the entire image for exclusion if the problematic regions are too extensive or critical to be simply masked out.

This process ensures that data compromised by non-Martian content or severe artifacts are meticulously identified and appropriately handled.

The direct outcome of this Grounded-SAM assisted semi-automated refinement is a rigorously curated collection of Martian stereo images. These images exhibit high visual integrity, characterized by predominantly unobstructed Martian surfaces, more balanced illumination across the scene, and a minimization of instrumental or environmental artifacts. Such a high-quality, clean dataset forms a reliable and robust foundation essential for the subsequent stages of metric-aware 3D reconstruction and, ultimately, for the training of generative simulation models like MarsGen.

### A.3   Details of M3arsSynth construction

**The challenge of conversion from CAHVOR to pinhole model**   This sections outlines the conversion from a CAHVOR ($C_{CAHVOR}, A_{CAHVOR}, H_{CAHVOR}, V_{CAHVOR}, O_{CAHVOR}, R_{CAHVOR}$ vectors) camera model to a pinhole model, highlighting the critical parameters and potential impediments. The conversion aims to derive pinhole model parameters (camera center $\mathbf{C}_{\text{pinhole}}$, rotation matrix $\mathbf{R}$, intrinsic matrix $\mathbf{K}$, and radial distortion coefficients $k_0, k_1, k_2$) from the CAHVOR parameters.

- **Camera Center:** $\mathbf{C}_{\text{pinhole}} = \mathbf{C}_{CAHVOR}$.

- **Rotation Matrix R:** Derived from normalized horizontal ($\mathbf{H}_n$) and vertical ($\mathbf{V}_n$) vectors, and the optical axis ($\mathbf{A}_{CAHVOR}$).

$$\mathbf{H}_n = (\mathbf{H}_{CAHVOR} - h_c \mathbf{A}_{CAHVOR})/h_s$$
$$\mathbf{V}_n = (\mathbf{V}_{CAHVOR} - v_c \mathbf{A}_{CAHVOR})/v_s$$
$$\mathbf{R} = \begin{pmatrix} \mathbf{H}_n^T \\ -\mathbf{V}_n^T \\ \mathbf{A}_{CAHVOR}^T \end{pmatrix}$$

- **Intrinsic Matrix K:** Determined by focal lengths ($f_u = h_s, f_v = v_s$) and principal point ($c_u = h_c, c_v = v_c$).

$$\mathbf{K} = \begin{pmatrix} h_s & 0 & h_c \\ 0 & v_s & v_c \\ 0 & 0 & 1 \end{pmatrix}$$

- **Radial Distortion** $k_0, k_1, k_2$**:** Calculated from $\mathbf{R}_{CAHVOR}$, with $k_1, k_2$ also depending on $h_s$.

$$k_0 = \mathbf{R}_{CAHVOR}[0]$$
$$k_1 = \mathbf{R}_{CAHVOR}[1]/(\text{pixel\_size} \times h_s)^2$$
$$k_2 = \mathbf{R}_{CAHVOR}[2]/(\text{pixel\_size} \times h_s)^4$$

The conversion critically depends on four scalar parameters:

- $h_s$: Horizontal focal length scaling factor.
- $v_s$: Vertical focal length scaling factor.
- $h_c$: Horizontal principal point coordinate.
- $v_c$: Vertical principal point coordinate.

If these four parameters $(h_s, v_s, h_c, v_c)$ are not available, the conversion to a pinhole model is not feasible. Therefore, these four scalar parameters are indispensable for a complete and accurate conversion from the CAHVOR to the pinhole model.

**3D Gaussian Splatting for Photorealistic Scene Modeling** We details key components and implementation specifics related to our 3D Gaussian Splatting (3DGS) model. The optimization of this model to accurately represent a 3D scene is driven by a combination of two primary loss functions: **Photometric Loss:** This loss ensures that the rendered images from the 3DGS representation closely match the input training images in terms of appearance. It penalizes differences in color and brightness, guiding the optimization towards visual fidelity.

$$\mathcal{L} = (1 - \lambda)\mathcal{L}_1 + \lambda\mathcal{L}_{\text{D-SSIM}} \tag{5}$$

With the goal of guiding the model into plausible geometry, we introduced a geometric prior loss in the process of model optimization , which is **Depth Regularization Loss**. This component utilizes the depth information output by the pre-trained large model and the depth obtained through the rasterization pipeline to supervise the geometric accuracy of the scene.

$$\mathcal{L}_{\text{depth}} = ||D_{render} - D^*_{\text{Metric Depth}}||_1 \tag{6}$$

The bilateral grid is a key component in addressing photometric variations and appearance inconsistencies in Martian stereo imagery, which arise from factors like differing camera hardware, lighting conditions, or ISP pipeline transformations. This per-view learnable function transforms the rendered output of a 3DGS model to better match the target image's appearance. Its implementation involves associating a 3D bilateral grid (a four-dimensional tensor $A \in \mathbb{R}^{W \times H \times D \times 12}$) with each training view, where $W, H$ represent spatial locations, $D$ represents pixel intensity values, and the final dimension stores parameters for a $3 \times 4$ affine color transformation matrix. For each rendered pixel, an affine transformation is retrieved via a differentiable slicing operation using trilinear interpolation based on its spatial location and a guidance intensity, allowing the grid's parameters to be learned end-to-end. Integrated into the 3DGS optimization, the grid processes rendered images before the photometric loss calculation, encouraging it to bridge appearance gaps, and a Total Variation loss is applied as smoothness regularization to prevent overfitting and encourage the modeling of low-frequency changes. The grid's resolution is typically much smaller than the input images ( we have selected here is $16 \times 16 \times 8 \times 12$ ) to ensure computational efficiency and focus on low-frequency variations, with adaptive sizing possible based on scene characteristics. This joint training approach mitigates appearance inconsistencies, leading to more photorealistic and geometrically consistent 3D scene representations.

**Camera Trajectory Synthesis and Motion Description** A cornerstone of generating diverse and informative video sequences for the M3arsSynth dataset is the meticulous design and execution of virtual camera trajectories. We begin by defining a repertoire of canonical camera trajectory types. These foundational trajectories, mathematically represented as a sequence of 6-Degrees-of-Freedom (6-DOF) poses $\mathcal{M}_{\text{traj}} = \{(R_t, T_t) \in \text{SE}(3) \mid t = 1, \ldots, N\}$, where $R_t$ signifies the camera's rotation matrix and $T_t$ denotes its translation vector at each discrete timestamp $t$, are engineered to encompass a wide spectrum of motion profiles. Our trajectory generation process begins by using the two camera poses, solved from the original stereo pair, as start and end keyframes. We then synthesize a variety

of smooth camera paths between these keyframes by applying predefined interpolation methods that follow canonical motion profiles. The spatial extent, or the overall scale and reach, of the predefined canonical trajectories is not fixed; instead, it is dynamically adjusted in response to the specific geometric characteristics of each individual 3D reconstructed Martian scene. This adaptation is primarily driven by the depth information derived from the reconstructed 3D model. Specifically, for regions within a scene that are identified as being in the near-field (characterized by relatively smaller depth values from the camera's perspective), the corresponding segments of the canonical trajectories are programmatically contracted or scaled down. Conversely, for regions designated as far-field (characterized by significantly larger depth values, indicating distant terrain elements or horizons), the trajectory segments are expanded or scaled up. This depth-adaptive scaling strategy is paramount for ensuring that the synthesized video data effectively and consistently covers the scene's content at appropriate levels of detail. Following the generation of these adaptively scaled trajectories, the precise 6-DOF pose parameters for each frame serve as the quantitative foundation from which natural language descriptions detailing the camera's motion characteristics are subsequently derived, forming a key component of the textual modality within the M3arsSynth dataset.

**Scene Content Captioning** The M3arsSynth dataset incorporates rich textual descriptions to enable and enhance multimodal learning. While depth and normal maps are directly derived from the 3D reconstructed scenes, and camera motion characteristics are derived from the 6-DOF pose parameters of the trajectories, the acquisition of descriptive scene content captions involves a sophisticated process leveraging a Vision Language Model (VLM). We generate scene content captions by applying a VLM, referenced as ChatGPT-4o, to selected views from the synthesized video sequences. Guided by expertise in Martian exploration from our authors, we developed structured prompts for the VLM to ensure high-quality annotations. These prompts include the input image, the classification of the Martian terrain depicted (e.g., "Regolith/Rocky Terrain", "Dunes/Ripples (Sand/Dust)", as shown in Figure 4 of the paper), and a basic descriptive outline of the scene. By conditioning the VLM with this structured input, we obtain detailed and contextually relevant textual descriptions of the visual content. We manually validated a random sample of 20% of the annotations and found no significant errors, which attests to the robustness of our pipeline. This methodology ensures that the textual modality is not only accurate but also aligned with the visual and geometric data, thereby creating a cohesive multimodal dataset suitable for training models like MarsGen for controllable video synthesis.

## A.4 MarsGen Architecture and Training Specifics

**Conditioning Mechanisms.** The MarsGen model integrates multimodal information through distinct conditioning pathways. Textual prompts are initially concatenated with video tokens; this combined representation is then processed through a global attention mechanism to achieve feature fusion. Camera trajectory information is incorporated by first representing camera poses using Plücker embeddings, which are subsequently injected into the model via a ControlNet architecture. Finally, initial video frames are conditioned by concatenating them with the input noise distribution, which then undergoes a denoising process to guide the generation.

**Fine-tuning Details.** The fine-tuning of MarsGen was conducted with the following hyperparameters. The learning rate was set to $1 \times 10^{-4}$. We utilized the AdamW optimizer. A cosine learning rate scheduler was employed, incorporating a warm-up phase. The batch size was configured to $1$ per GPU. Training was performed for $8,000$ steps, with gradient accumulation implemented over $2$ steps.

## B  Additional Experimental Results and Analysis

### B.1  Qualitative Comparisons of Video Generation

The main paper presented quantitative comparisons of our generator against image-to-video and camera-controlled image-to-video models. This appendix provides additional quantitative results against other video generation models. Sora and ViewCrafter, for instance, evidently lack specialized modeling for dynamic Martian scenes, leading to uncontrollable video sequences inconsistent with the theme. This further validates the significance of our proposed dataset.

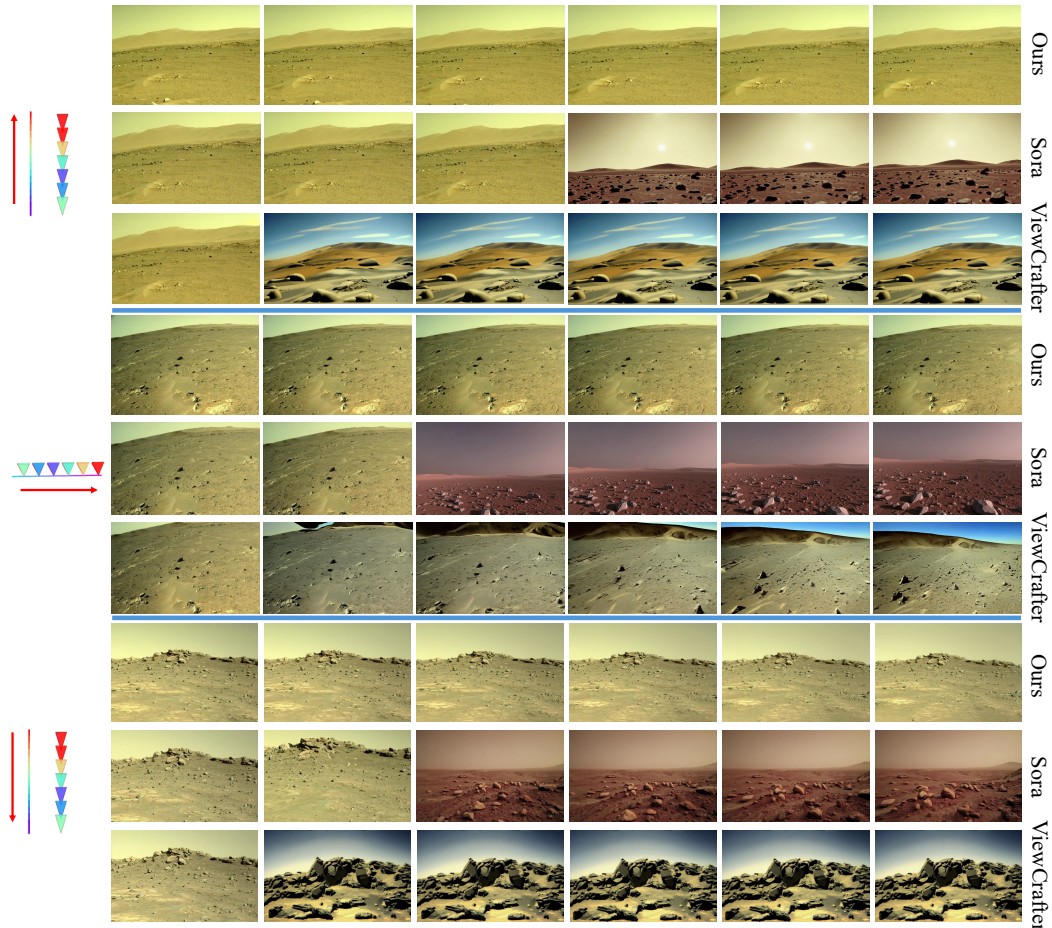

Figure A: **Qualitative comparison of video generation models on dynamic Martian scenes.** Each group of image sequences compares our model against Sora and ViewCrafter under a specific camera control condition. Our model demonstrates improved coherence with the intended camera control and greater thematic consistency for Martian landscapes, whereas Sora and ViewCrafter occasionally produce less controlled or thematically divergent outputs. The relevant comparison video files (in .mp4 format) are located in the corresponding subdirectories under the `comparison/` folder. For example, Sora's examples are in the `comparison/sora/` directory,

For more comparisons of videos generated by our MarsGen model against the ground truth (GT), please refer to the `ours/` folder. Examples of other models' failures in generating dynamic Martian scenes can be found in the `others/` folder.

## B.2  Qualitative Comparisons of 3D Reconstruction Pipelines

Fig. B illustrates a qualitative comparison of 3D reconstruction outputs from different methodologies when applied to demanding Martian stereo image pairs. The top row of the figure presents results achieved by our M3arsSynth pipeline, which consistently demonstrates the ability to generate coherent and detailed 3D reconstructions of the Martian terrain. These outputs effectively capture the complex geometry and features of the landscape.

In contrast, the second row of Fig. B displays reconstructions produced by the MASt3R pipeline. As indicated by the highlighted regions within the red boxes, MASt3R can encounter difficulties, leading to potential inaccuracies, loss of fine details, or artifacts. While MASt3R achieves 100% data utilization in quantitative assessments, it exhibits a significantly high reprojection error (46.98 px according to Table 2), which may stem from overfitting to unreliable depth priors.

Further highlighting the difficulties faced by existing techniques, traditional Structure-from-Motion (SfM) pipelines like COLMAP demonstrate extremely low utilization and robustness when processing Martian data. For the specific visual examples shown in Fig. B, the COLMAP pipeline failed to generate a usable reconstruction in every instance, indicating its poor suitability for these challenging datasets. This qualitative observation is strongly supported by quantitative data presented in Table 2 of the main paper, which shows that COLMAP experiences failures on nearly 30% of preprocessed Martian image pairs. Such a high failure rate severely restricts its practical application for comprehensive 3D modeling of Martian environments.

In stark contrast, our proposed M3arsSynth pipeline achieves 100% data utilization and successfully reconstructs dense point clouds (averaging 250,000 points) while maintaining a competitive reprojection accuracy (0.77 px, as detailed in Table 2). This robust performance, delivering both high data utilization and superior reconstruction quality, underscores the efficacy of our M3arsSynth data engine in producing the reliable and accurate 3D models that are essential for creating high-fidelity simulations and facilitating advanced video synthesis of Martian terrains.

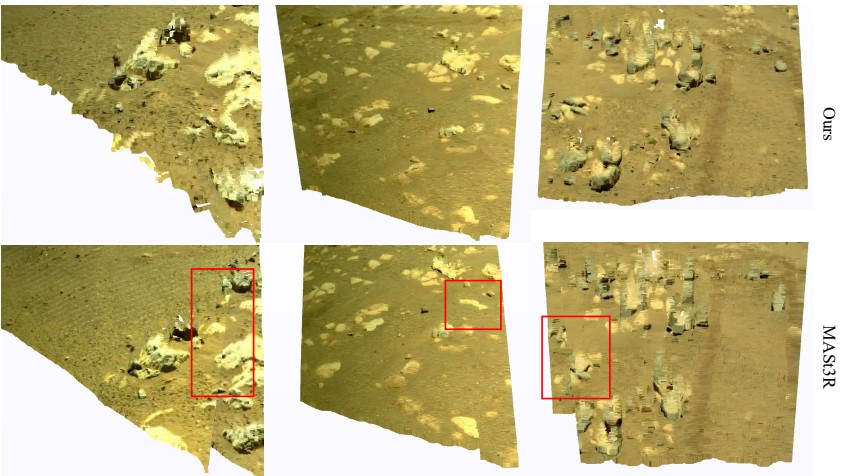

Figure B: **Qualitative comparison of 3D reconstruction pipelines on challenging Martian stereo imagery.** The top row displays reconstructions from our M3arsSynth pipeline, while the bottom row shows results from MASt3R, with red boxes highlighting areas of potential inaccuracy or detail loss. Notably, the COLMAP pipeline failed to produce reconstructions for all depicted examples, underscoring its limitations in robustness and data utilization on Martian datasets.

### B.3 Discussion on Geometric Initialization Component Choices

Our experiments, summarized in Table A, validate our component choices for geometric initialization. Our primary goal is accurate, metric-scale reconstruction, which is critical for Martian terrain analysis but not a direct output of methods like VGGT which predicts normalized depth.

The ablation study highlights the limitations of alternatives. The full `MASt3R(full)` pipeline and its intrinsic estimation (`Ours(MASt3R intrin.)`) are unstable (46.980 px and 2.016 px, respectively) due to inaccurate 2D correspondences. In contrast, we found VGGT provides more stable intrinsic estimations, likely due to its DINOv2-initialized encoder and normalized training targets.

Furthermore, using VGGT's outputs directly is suboptimal. Scaling its normalized depth to metric scale (`VGGT(metric depth)`) is also suboptimal (1.748 px). Using its predicted extrinsics (`Ours(VGGT extrin.)`) also yields poor results (2.113 px). This is because these non-metric scale poses are fundamentally incompatible with the metric depth priors from Metric3D v2, which are essential to our pipeline. Our approach successfully integrates the strengths of these models while ensuring metric-scale consistency.

Table A: Ablation study on geometric initialization components. Our full method achieves the lowest 2D reprojection error, indicating the most accurate geometric setup.

|  | MASt3R(full) | VGGT(metric depth) | Ours(MASt3R intrin.) | Ours(VGGT extrin.) | **Ours** |
|---|---|---|---|---|---|
| 2D Reproj. ↓ | 46.980 | 1.748 | 2.016 | 2.113 | **0.770** |

## B.4 Discussion on Generalization to Lunar and Terrestrial Environments

While our primary motivation stems from Martian exploration, our method is designed for planetary scenes with sparse views and low-texture surfaces. We evaluated our pipeline on similar environments, including public lunar data, performing novel view synthesis from two views.

Furthermore, to test robustness in a general sparse-view context, we performed novel view synthesis on the terrestrial RealEstate10K dataset using only two-view inputs. We benchmarked our approach against Splatt3R and InstantSplat.

The preliminary results, shown in Table B, demonstrate our method's effectiveness in these challenging sparse-view conditions on both lunar and terrestrial data.

Table B: Sparse-view novel view synthesis benchmarks on Lunar and RealEstate10K data.

| Dataset | Method | PSNR ↑ | SSIM ↑ | LPIPS ↓ |
|---|---|---|---|---|
| Lunar | Ours | 29.60 | 0.920 | 0.213 |
| RealEstate10K | Splatt3R | 15.11 | 0.492 | 0.442 |
|  | InstantSplat | 19.64 | 0.560 | 0.291 |
|  | **Ours** | **20.81** | **0.717** | **0.264** |

