# OpenReview forum: "Martian World Model: Controllable Video Synthesis with Physically Accurate 3D Reconstructions"
_NeurIPS.cc/2025/Datasets_and_Benchmarks_Track — NeurIPS 2025 Datasets and Benchmarks Track poster_

### Official Review · Reviewer_cFSn · 2025-06-27

**Rating:** 4
**Confidence:** 3

**Summary:**

This paper introduces a novel 3D video datasets for martian landscape. The dataset is constructed by curating stereo navigation images from MASA’s Planetary Data System and then reconstruct the images with geometrical foundation models to get additional geometrical information, like cameras, depths and etc. The reconstructed information is then rendered into a denser datasets which could be used to train video generation models which could synthesize martian videos with camera control ability.

**Dataset Code Accessibility:**

Yes

**Ethical Considerations:**

No, there are no or only very minor ethics concerns

**Final Justification:**

My concerns have been well addressed in the rebuttal, therefore I vote to acceptance.

**Limitations Weaknesses:**

The authors claim that the datasets are rendered from the reconstructed 3D gaussian splatting. I’m curious about this:
* How to evaluate the reconstruction quality of these 3D gs? If the reconstruction contains some errors, then the rendered images could not be used.
* How to select a proper camera trajectory for rendering? There should be some methods to ensure that the rendered images do not contain any occluded or low confident areas.
* There could be a comparison between adopting rendered images and adopting the original images for training the video generation models.

**Strengths Contributions:**

* Geometrical foundation models are growing rapidly. Adopting geometrical foundation models to facilitate the construction of 3D datasets is a promising and interesting direction.
* MarsGen trained on the proposed dataset shows better performance on the martian video generation compared to other existing video diffusion methods.
* The paper is well-written and easy to understand.

---

> ### Author Rebuttal · Authors · 2025-07-31
>
> ## Author Response to **Reviewer cFSn**
>
> Thank you for the constructive feedback and for finding our paper well-written. We are especially glad you found our direction of using geometric foundation models to construct 3D datasets to be both promising and interesting. Like you, we believe this is a key approach for creating specialized, high-quality data. In the following, we address your other comments and provide further clarifications.
>
> ### **Q1: Evaluation of the reconstruction quality**
>
> To quantitatively evaluate  our 3D reconstructions, we use the 2D reprojection error. This metric measures the pixel distance between an observed point in the original image and its corresponding 3D point when re-projected back onto the image using the estimated camera parameters. It thereby jointly assesses the geometric accuracy of the 3D model and the estimated camera pose. The results are reported in Table 2 of our main draft.
>
> Evaluating novel view synthesis quality is infeasible as our methodology reconstructs scenes from sparse stereo pairs (just 2 images used for training the 3D-GS model), leaving no additional ground truth views for quantitative comparison. Therefore, we use the traditional reprojection error, a robust metric for few-view scenarios. It measures the pixel distance between a projected 3D point and its corresponding 2D image point.
>
> ---
>
> ### **Q2: The selection of the rendered camera trajectory**
>
> Our trajectory generation process begins by using the two camera poses, solved from the original stereo pair, as keyframes. We then synthesize a variety of smooth camera paths between these keyframes by applying predefined interpolation methods that follow canonical motion profiles. To ensure the path is coherent and avoids collisions, we employ depth-adaptive scaling. This technique leverages the depth information to adjust the trajectory's scale. Specifically, trajectories are contracted for near-field regions and expanded for far-field regions for wider movements.
>
> ---
>
> ### **Q3: Sparse-View data is insufficient for Martian world model training**
>
> Using the original sparse data directly constitutes a sparse (just 2) view generation task, not one of continuous video generation. This distinction is critical, as our goal of building a video-based world model for dynamic, continuous simulation[1] cannot be met by a model trained only on the original two-view images.
>
> This justification validates that our M3arsSynth pipeline is the essential prerequisite that solves this fundamental data challenge. It reconstructs the sparse source images to 3D models and renders dense, continuous video clips necessary to train a physically plausible world model (a.k.a., MarsGen).
>
> ---
>
> ### **Summary**
>
> Thank you again for your valuable feedback and constructive comments. In our responses, we have aimed to clarify the rationale behind our key methodological choices. We have explained how we evaluate reconstruction quality under sparse-view constraints, our depth-guided process for generating safe camera trajectories, and why our data-to-video pipeline is essential for building a true world model.
>
> We have aimed to provide thorough explanations for all points raised. We would be very grateful if you would take these detailed responses and clarifications into account when finalizing your review. Please do not hesitate to let us know if we can be of further assistance.
>
> ---
>
> ### **References**
> [1] Jingtao Ding, Yunke Zhang, Yu Shang, Yuheng Zhang, Zefang Zong, Jie Feng, Yuan Yuan, Hongyuan Su, Nian Li, Nicholas Sukiennik, et al. Understanding world or predicting future? a comprehensive survey of world models. *ACM Computing Surveys*, 2024.

---

### Official Review · Reviewer_Ehck · 2025-06-30

**Rating:** 5
**Confidence:** 3

**Summary:**

This paper introduces an integrated framework for synthesizing realistic Martian landscape videos to address critical challenges in mission rehearsal and robotic simulation. The proposed solution comprises two innovations: 1) The M3arsSynth data engine transforms sparse, photometrically inconsistent stereo navigation imagery from NASA's Planetary Data System into physically accurate 3D environments. The engine generates a comprehensive multimodal dataset featuring rendered video sequences, depth/normal maps, camera trajectories, and text descriptions. 2) The MarsGen video generator synthesizes novel, geometrically consistent Martian terrain videos conditioned on initial image frames, camera trajectories, or text prompts. Experimental validation demonstrates significant advancements over previous video synthesis models.

**Dataset Code Accessibility:**

Yes

**Ethical Considerations:**

No, there are no or only very minor ethics concerns

**Final Justification:**

1. The missing dataset license and dataset card have been added.

2. The explanation for not using VGGT for camera extrinsics (due to its lack of metric scale, critical for physical accuracy in Martian terrain modeling) is well-supported by experimental results.

3. The commitment to incorporating relighting in future work to enhance dataset diversity and model generalization directly addresses concerns about limited environmental variation, providing a clear path to improve generalization.

**Limitations Weaknesses:**

1. Missing license and dataset card.
2. I notice that the paper uses VGGT to predict camera intrinsics while employing a PnP solver to estimate relative camera extrinsics. I'm curious why VGGT isn't also used for camera extrinsics. While the authors mention that directly initializing 3DGS with point clouds generated from VGGT can produce artifacts, I think the camera extrinsics estimated by VGGT could still be potentially useful.
3. The study relies exclusively on NASA datasets without applying data augmentation techniques (e.g., lighting variations), limiting environmental diversity. This restriction potentially compromises model generalization for dynamic Martian conditions like low-light navigation.

**Strengths Contributions:**

1. By leveraging geometric foundation models, this pipeline overcomes textureless terrain and sparse viewpoints, reconstructing metric-scale Martian surface models.
2. Generating the first multimodal Mars dataset with synchronized video, depth maps, and text descriptions.
3. Enables controllable video generation using a controlnet manner, starting from a single-view image and conditioned on camera poses or text prompt.

---

> ### Author Rebuttal · Authors · 2025-07-31
>
> ## Author Response to **Reviewer Ehck**
>
> Thank you for the thorough review and for highlighting our key contributions, particularly in overcoming textureless terrain and sparse viewpoints to create an accurate 3D environment and enabling controllable video generation. We agree that these capabilities are critical for realistic robotic simulation. In what follows, we clarify key points and address your comments in detail.
>
> ### **Q1: Missing License and dataset card**
>
> We have now added the license and the dataset card to our dataset on the Hugging Face Hub as you suggested. We appreciate your reminder.
>
> ---
>
> ### **Q2: VGGT extrinsics comparison**
>
> The primary reason we do not use the camera extrinsics predicted by VGGT is that they lack a metric scale. These non-metric scale poses are incompatible with the metric depth priors that are fundamental to our pipeline. Physical accuracy is crucial for Mars exploration to enable a true-to-scale understanding of its terrain.
>
> To demonstrate the impact of this incompatibility, we did conduct an experiment using VGGT's predicted extrinsics. As the results in the table below show, this approach (noted as Ours(VGGT extrin.))  leads to inferior performance.
>
> |                  | MASt3R(full) | Ours(VGGT extrin.) |   Ours    |
> | ---------------- |:------------:|:------------------:|:---------:|
> | 2D Reroj. &darr;  |    46.980    |       2.113        | **0.770** |
>
> ---
>
> ### **Q3: Data augmentation techniques**
>
> Applying data augmentation to the source images would disrupt the multi-view consistency required for reliable geometric correspondence matching, likely causing the reconstruction to fail.
> We agree with the suggestion to apply augmentation on the reconstructed scenes before rendering the video. The ability to perform relighting on our high-quality 3D models to generate more diverse training data is a promising direction [1], and we will include this in our discussion of future work.
>
> ---
>
> ### **Summary**
> Thank you again for your valuable feedback. In response, we have added the requested dataset card and license. We have also clarified our technical choices regarding the necessity of metric-scale reconstruction and our principled approach to data augmentation. Your suggestion for post-reconstruction augmentation is highly appreciated and has been included in our future work.
>
> We hope these clarifications and improvements have fully addressed your concerns and would be very grateful if you would consider them in your final assessment. Please let us know if any further questions remain.
>
> ---
>
> ### **References**
>
> [1] Haiyang Bai, Jiaqi Zhu, Songru Jiang, Wei Huang, Tao Lu, Yuanqi Li, Jie Guo, Runze Fu, Yanwen Guo, Lijun Chen. GaRe: Relightable 3D Gaussian Splatting for Outdoor Scenes from Unconstrained Photo Collections. *arXiv preprint arXiv:2507.20512*, 2025.

---

> > ### Comment · Reviewer_Ehck · 2025-08-03
> > **Thanks for the detailed response.**
> >
> > Thanks for the detailed response. The explanation that VGGT is not used for camera extrinsics due to its lack of metric scale is reasonable and addresses my curiosity on this point. I am also looking forward to the future work incorporating relighting to enhance the dataset’s diversity and the model’s generalization. With these clarifications, my concerns have been resolved, and I am willing to raise the rating.

---

> > > ### Author Response · Authors · 2025-08-05
> > > **Thank You for Your Support and Constructive Feedback**
> > >
> > > Dear Reviewer Ehck,
> > >
> > > Thank you for the positive follow-up. We are glad our explanation regarding camera extrinsics was helpful.
> > >
> > > We also appreciate your valuable suggestion of using relighting on our 3D models to enhance data diversity. We will incorporate this discussion into the Conclusion section of our revised draft.
> > >
> > > Thank you again for raising your rating and for the constructive feedback that we believe would strengthen our manuscript.

---

### Official Review · Reviewer_cDFy · 2025-07-05

**Rating:** 4
**Confidence:** 4

**Summary:**

The M3arsSynth dataset offers significant research value and practical significance in the field of controllable video synthesis and physically accurate 3D reconstruction for the Martian environment. It provides researchers with a high-quality, multimodal data resource and a robust benchmarking tool. However, it also has certain limitations, such as a limited dataset scale, incomplete modalities, and high technical barriers. Addressing these shortcomings would better meet researchers' needs and further advance related research fields.

**Additional Feedback:**

The dataset is primarily designed for controllable video synthesis and 3D reconstruction in the Martian environment, which is quite limited in terms of scenarios. Such topic should be investigated in multiple cases

**Dataset Code Accessibility:**

Yes

**Ethical Considerations:**

No, there are no or only very minor ethics concerns

**Final Justification:**

Towards acceptance with respect to the rebuttal

**Limitations Weaknesses:**

- 1. *Limited Dataset Scale*: The dataset may have a relatively small number of scenes and video sequences, which could limit the generalization and robustness of algorithms trained on it. Expanding the dataset scale to include more Martian scenes and video sequences would enhance its value for research.
- 2. *High Technical Barriers for Data Processing*: The dataset's construction involves complex data processing techniques, such as CAHVOR to pinhole model conversion, 3D Gaussian Splatting optimization, and camera trajectory synthesis. These require researchers to possess extensive expertise in computer vision and related fields, potentially posing technical challenges for some users.
- 3. *Potential Annotation Errors*: Although the dataset undergoes rigorous filtering and preprocessing, there may still be some annotation errors or inaccuracies. For example, the textual descriptions generated by the vision-language model may not perfectly align with the actual scene content. These errors could affect the reliability of algorithm evaluations.
- 4. *Limited Applicability*: The dataset is primarily designed for controllable video synthesis and 3D reconstruction in the Martian environment, limiting its applicability to other planetary or terrestrial scenarios. Its value and utility in broader research contexts may be constrained.

**Strengths Contributions:**

- 1. *Unique Dataset Theme*: M3arsSynth focuses on the Martian environment, offering a dataset for controllable video synthesis and physically accurate 3D reconstruction in this unique planetary context. It provides researchers with valuable data resources for exploring video synthesis and 3D reconstruction in extraterrestrial environments, filling a gap in the field of planetary science and computer vision research.
- 2. *Diverse and High-Quality Data*: The dataset includes a variety of data modalities, such as images, 3D reconstructions, camera trajectories, and textual descriptions. The images undergo rigorous automated and semi-automated filtering and preprocessing to ensure high quality. The 3D reconstructions are generated using advanced 3D Gaussian Splatting techniques, offering high geometric accuracy and photorealism. The camera trajectories are meticulously designed to cover different motion profiles, and the textual descriptions are generated through a combination of vision-language models, providing rich semantic information.
- 3. *Robust Data Processing Pipeline*: The dataset employs a sophisticated data processing pipeline, including automated data filtering strategies such as removing low-quality thumbnails and grayscale images, eliminating redundant content via perceptual hashing, excluding blurry and low-sharpness images, and filtering out visually unusable frames. This ensures the dataset's semantic diversity and high visual integrity. The semi-automated refinement stage, assisted by Grounded-SAM, further enhances the dataset's quality by addressing complex visual challenges like partial occlusions and subtle lens distortions.
- 4. *Support for Multimodal Learning*: The integration of images, 3D reconstructions, camera trajectories, and textual descriptions enables multimodal learning. Researchers can leverage the correlations and complementary information between different modalities to develop more powerful models for controllable video synthesis and 3D reconstruction, driving the advancement of multimodal learning research.
- 5. *Accurate 3D Reconstruction and Video Synthesis*: The M3arsSynth pipeline achieves 100% data utilization and successfully reconstructs dense point clouds while maintaining competitive reprojection accuracy. The generated videos exhibit high coherence with camera controls and strong thematic consistency, demonstrating the dataset's effectiveness in supporting accurate 3D reconstruction and high-quality video synthesis.

---

> ### Author Rebuttal · Authors · 2025-07-31
>
> ## Author Response to **Reviewer cDFy**
>
> We appreciate the reviewer's detailed feedback and for highlighting the strengths of our work, particularly the robust data processing pipeline and the resulting high-quality, multimodal data. We believe this rigorous approach is essential for creating reliable benchmarks. In the following sections, we will address the noted limitations and provide further clarification.
>
> ### **Q1: Limited dataset scale**
>
> Thank you for pointing this out. The raw data we used is the entire available dataset from the NASA Planetary Data System (PDS). Due to the inherent difficulties of deep space exploration, such as severe bandwidth and equipment constraints, this data is known to be noisy and limited in usable scale. Therefore, while the total raw collection is vast, we deliberately filtered it to produce a clean, high-quality dataset for our experiments. We believe a high-quality dataset is far more valuable than a larger, noisier one.
>
> ---
>
> ### **Q2: High technical barriers for data processing**
>
> We thank the reviewer for highlighting this. A core objective of our work is precisely to abstract this complexity away from the end-user. By handling the entire data processing pipeline, we provide the research community with a ready-to-use, high-quality multimodal dataset. This allows researchers to focus on downstream applications without needing specialized expertise in 3D vision or camera geometry. Furthermore, we will release our code, including a user-friendly single script to run the pipeline, which will allow for its use without requiring in-depth knowledge of the technical details.
>
> ---
>
> ### **Q3: Potential annotation errors**
>
> We acknowledge that ensuring annotation accuracy is a general challenge for both planetary and Earth-based data. Specifically for the textual descriptions, our pipeline is in line with recent methodologies[1, 2] and strictly follows the well-established expert-in-the-loop principle. Guided by expertise in Martian exploration from our authors, we developed structured prompts for the VLM to ensure high-quality annotations. We manually validated a random sample of 20% of the annotations and found no significant errors, which attests to the robustness of our pipeline. We will add details about this verification process in our revised version.
>
> ---
>
> ### **Q4: Limited applicability**
>
> While our primary motivation stems from Martian exploration, our method is fundamentally designed for planetary scenes characterized by sparse views and low-texture surfaces. We tested our pipeline on similar planetary environments, like public lunar data sourced from internet videos, performing novel view synthesis from two views, with the final quality evaluated against a held-out ground truth image. However, general-purpose methods often struggle in these conditions. As shown in Table 2 of our paper, other approaches like MASt3R exhibit high reconstruction errors on such challenging data.
>
> |  | PSNR &uarr; | SSIM &uarr; |  LPIPS &darr; |
> | -------- | ---- | ---- | --- |
> | Lunar data    | 29.60 | 0.920 |  0.213   |
>
> Furthermore, to test the method's robustness in a general sparse-view context, we performed novel view synthesis on the terrestrial RealEstate10K dataset using only two-view inputs. We benchmarked our approach against two leading methods: Splatt3R, a feedforward sparse-view reconstruction method, and InstantSplat , an optimization-based approach for pose-free GS. The preliminary results are presented below:
>
> |              | PSNR &uarr; | SSIM &uarr; | LPIPS &darr; |
> | ------------ | -------- | -------- | --- |
> | Splatt3R     | 15.11    | 0.492    |0.442 |
> | InstantSplat | 19.64   | 0.560   | 0.291 |
> | Ours         | **20.81**    | **0.717**   | **0.264** |
>
> While designed for planetary environments, these results indicate that our approach remains competitive in general sparse-view reconstruction scenarios.
>
> ---
>
> ### **References**
> [1] Ang Wang, Baole Ai, Bin Wen, Chaojie Mao, Chen-Wei Xie, Di Chen, Feiwu Yu, Haiming Zhao, Jianxiao Yang, Jianyuan Zeng, et al. Wan: Open and advanced large-scale video generative models. *arXiv preprint arXiv:2503.20314*, 2025.
>
> [2] Yoav HaCohen, Nisan Chiprut, Benny Brazowski, Daniel Shalem, Dudu Moshe, Eitan Richardson, Eran Levin, Guy Shiran, Nir Zabari, Ori Gordon, Poriya Panet, Sapir Weissbuch, Victor Kulikov, Yaki Bitterman, Zeev Melumian, and Ofir Bibi. Ltx-video: Realtime video latent diffusion. *arXiv preprint arXiv:2501.00103*, 2024

---

> > ### Comment · Area_Chair_wK9t · 2025-08-05
> >
> > Dear reviewer cDFy,
> >
> > Could the rebuttal have addressed your concerns? Please join the discussion and update your score.
> >
> > The AC

---

> > ### Comment · Reviewer_cDFy · 2025-08-05
> >
> > The updated ablation with MASt3R and Splatt3R makes the work more applicable. I would like to raise the score accordingly

---

### Official Review · Reviewer_xeWQ · 2025-07-09

**Rating:** 5
**Confidence:** 3

**Summary:**

In this paper, the authors introduce an approach for synthesizing realistic Martian landscape videos. It begins with M3arsSynth, a data curation framework designed to process sparse stereo navigation images from NASA’s PDS, resulting in a multimodal Mars dataset that includes high-fidelity 3D surface models. Building on this, MarsGen, a video-based Martian terrain generator, is developed and fine-tuned using the M3arsSynth dataset to produce novel, controllable, and 3D-consistent video sequences. Experimental results demonstrate that the proposed method surpasses existing Earth-trained video synthesis models in visual fidelity, 3D consistency, and camera controllability.

**Dataset Code Accessibility:**

No

**Dataset Code Comments:**

The reviewer can not find the code link. The authors mentioned "We just promise release our dataset" in the paper. They did not mention that the code will be publicly available.

**Ethical Considerations:**

No, there are no or only very minor ethics concerns

**Final Justification:**

The authors have addressed my concerns. I believe this work would be more solid if the full codebase were also open-sourced.

**Limitations Weaknesses:**

-- The title "Martian World Models" may be misleading. In fact, only a single video generation model is developed in this paper, rather than multiple models as the title might suggest.

-- For depth alignment, shouldn't the depths from two views first be transformed into a common view before applying the alignment procedure of Eq. (2)? Directly aligning depths from different views using the least squares formula seems to be incorrect.

-- The authors mentioned that the bilateral grid is integrated into 3DGS. However, how it is integrated is unclear. In the supplementary material, although the  bilateral grid is introduced, how it interacts with 3DGS is still missing.

-- How are the conditions encoded and injected into the ControlNet-based VDiT? Please provide a detailed structural diagram to illustrate the process.

-- The authors employed VGGT to obtain the camera intrinsic parameters. However, why did they only compare the geometric initialization method using MASt3R in the 3D reconstruction experiments? The point cloud quality from VGGT is lower than that of MASt3R. Would using the intrinsic parameters derived solely from MASt3R result in better outcomes compared to those obtained from VGGT?


Minor:
- L108-L110: Sec. xxx should be followed by the verb in its third-person singular form.

**Strengths Contributions:**

+ A M3arsSynth data engine is developed. It processes NASA stereo navigation imagery into a versatile multimodal Mars dataset. By leveraging geometric foundation models, it creates metric-scale 3D environments, effectively addressing issues like sparse-view coverage and photometric inconsistencies to produce over 10K physically accurate 3D Martian surface models.

+ A MarsGen video generator is proposed, which, based on the data produced by M3arsSynth, can generate photorealistic and 3D-consistent video sequences of Martian terrains from single-view image inputs, camera poses, or textual prompts. This approach significantly outperforms models primarily trained on terrestrial data.

---

> ### Author Rebuttal · Authors · 2025-07-31
>
> ## Author Response to **Reviewer xeWQ**
>
> We thank the reviewer for your insightful feedback and for recognizing our core contribution: the M3arsSynth engine for creating physically-accurate data and the subsequent high-performance MarsGen video generator. We agree that a high-quality, domain-specific dataset is the crucial foundation for robust generative world models. Below, we address the other points raised and provide further clarifications.
>
> ### **Q1: Suggestions for revising the title**
> We will revise the title to "Martian World Model" to more accurately reflect the content of the paper.
>
> ---
>
> ### **Q2: Clarification on transformation in the depth alignment**
>
> We thank the reviewer for pointing out this important discrepancy. We made a typo  in Sec. 3.2 5th paragraph. Our implementation for the alignment process is as you described. For your reference, below is the pseudo-code for this procedure.
>
> First, we back-project the depth map $D\_0$ from view 0 to reconstruct its corresponding 3D point cloud $P_0$. We then project it onto view 1's image plane. This warping process yields a new sparse depth map $D'\_{0\to1}$ and a Boolean mask $M\_{sparse}$, indicating valid projection areas in view 1. The depth value $d'\_1$ is stored in $D'\_{0\to1}$ at the projected pixel coordinates $(u'\_1/d'\_1, v'\_1/d'\_1)$.
> $$P_0 = D_0(p_0) \cdot K_0^{-1} \begin{bmatrix} u_0 \\ v_0 \\ 1 \end{bmatrix}^T$$ $$\begin{bmatrix} u'_1 \\ v'_1 \\ d'_1 \end{bmatrix}^T = K_1 (R P_0 + t)$$
>
> Then, we align the original monocular depth map of view 1 $D\_1$, with the warped sparse depth map $D'\_{0\to1}$. We solve a least-squares regression problem only within the valid region defined by the mask $M\_{sparse}$.
> $$\min_{s,b}\sum_{(u,v) \in M_{sparse}}(s\cdot D_{1}(u,v)+b-D'_{0\to1}(u,v))^{2}$$
>
> We will revise Sec. 3.2 and the corresponding Eq.(2) to reflect this implementation.
>
> ---
>
> ### **Q3: The integration of the bilateral grid with Gaussian Splatting**
>
> We will add more detail about how to integrate the bilateral grid into 3D GS:
> Specifically, our implementation is based on gsplat[1]. We apply a per-view 3D bilateral grid as a differentiable post-processing layer to the rendered image, which models view-dependent effects. This grid is jointly optimized with the Gaussian parameters by minimizing the difference between the post-processed render and the corresponding training view, while a total variation loss is used to regularize the grid for smoothness. And we will revise Sec. 3.2 to properly cite this work gsplat[1].
>
> ---
>
> ### **Q4: Explanation of MarsGen conditional control mechanism**
>
> We have described this mechanism in the first paragraph of Appendix A.2. Here we draw a simple structure diagram with additional details:
> - **Text prompt**: Text is tokenized into embeddings, which are then concatenated with the sequence of video patch tokens. Global self-attention is subsequently applied to this combined sequence.
> - **Initial Frame**: The initial video frame is conditioned by concatenating it with the input noise distribution.
> - For **camera trajectory conditioning**, we first project the camera poses into Plücker embeddings. These embeddings are then processed by a ControlNet-like architecture composed of smaller VDiT blocks, utilizing a hidden dimensionality of 128 and 4 attention heads. Finally, the output of this conditioning network is integrated with video tokens by performing a summation in each main DiT block.
>
> For a more detailed visual illustration of a similar architecture, we refer the reviewer to the structural diagram in AC3D [2].
>
> (please paste this diagram to mermaid live editor)
> ```mermaid
> graph TD
>     subgraph "Inputs"
>         direction LR
>         A[Text]
>         I[Init Frame]
>         B[Noise]
>         C[Camera]
>     end
>     subgraph "VDiT Blocks"
>         Concat1(Concatenate)
>         D(Global-Attention)
>         E(Summation +)
>
>         Concat1 --> D
>         D --> E
>     end
>     subgraph "ControlNet"
>         Concat2(Concatenate)
>         G[ControlNet Blocks]
>         Concat2 --> G
>     end
>     A -- "Encode Prompt" --> Concat1
>     B -- "Input" --> Concat1
>     I -- "VAE Encode" --> Concat1
>     B -- "Input" --> Concat2
>     C -- "Plücker Embedding" --> Concat2
>     G -- " " --> E
>     E -- "Final Output" --> H(Generated Video)
> ```
>
> ---
>
> ### **Q5: Geometric initialization method and MASt3R intrinsics**
> Our experiment validates that our method offers superior geometric initialization compared to VGGT geometry initialization, noted as VGGT(metric depth) in the table below.
>
> We compared MASt3R instead of VGGT geometry initialization because VGGT predicts normalized depth instead of metric-scale reconstruction that our approach targets. We prioritize metric scale because it is crucial for space exploration applications like mission planning and terrain analysis. We investigated VGGT geometry initialization, where we scale its normalized depth prediction to a metric scale and then solve for the camera extrinsics using a PnP solver.
>
> This performance advantage stems from our integration of more robust, specialized foundation models. Metric3D v2 was trained on over 16 million images from thousands of different camera models. This enables exceptional zero-shot generalization and the ability to predict accurate metric scale depth. The training data for VGGT and MASt3R are both on the scale of several million images. Image matching is a robust task for low-textured images due to its mechanism being based on local feature correspondence. The integration of these two specialized models provides a more robust and accurate framework for processing Martian data.
>
> Using MASt3R for intrinsic estimation (noted as Ours(MASt3R intrin.) ) proves unstable in our Martian surface scenarios as its underlying 2D correspondences can be inaccurate (as shown in table below, noted as MASt3R(full)). Moreover, MASt3R’s  iterative approach is time-consuming, which conflicts with our goal of  creating an efficient data engine.
> In contrast, our empirical experiments find that VGGT provides better intrinsic estimations. We speculate this is because VGGT uses a DINOv2-initialized encoder and is trained on normalized targets. This approach appears to result in better convergence, a point discussed by the authors of VGGT in issue #56 in their GitHub repo.
>
> |           | MASt3R(full) | VGGT(metric depth) | Ours(MASt3R intrin.) | Ours      |
> | --------- |:------------ |:-------------------:| ----------------------- | --------- |
> | 2D Reroj. &darr; | 46.980       |        1.748        | 2.016                   | **0.770** |
>
> ---
>
> ### **Q6. L108-L110: Sec. xxx should be followed by the verb in its third-person singular form**
>
> We will fix the typo. Thanks.
>
> ---
>
> ### **Response to Dataset Code Comments**
> As indicated in our response to the NeurIPS checklist (specifically item 5), we initially provided only the dataset for review purposes. We will release our entire codebase before the end of the review stage.
>
> ---
>
> ### **References**
> [1] Vickie Ye, Ruilong Li, Justin Kerr, Matias Turkulainen, Brent Yi, Zhuoyang Pan, Otto Seiskari, Jianbo Ye, Jeffrey Hu, Matthew Tancik, and Angjoo Kanazawa. gsplat: An open-source library for gaussian splatting. *Journal of Machine Learning Research*, 26(34):1–17, 2025.
>
> [2] Sherwin Bahmani, Ivan Skorokhodov, Guocheng Qian, Aliaksandr Siarohin, Willi Menapace, Andrea Tagliasacchi, David B Lindell, and Sergey Tulyakov. Ac3d: Analyzing and improving 3d camera control in video diffusion transformers. *arXiv preprint arXiv:2411.18673*, 2024.

---

> > ### Comment · Reviewer_xeWQ · 2025-08-05
> >
> > The rebuttal has addressed my concerns. I believe this work would be more solid if the full codebase is open-sourced.

---

> > > ### Author Response · Authors · 2025-08-08
> > >
> > > Dear Reviewer xeWQ,
> > >
> > > Thank you for your valuable suggestions. In response, we have made several key updates to strengthen our work:
> > >
> > > - We have now open-sourced the full codebase.
> > > - We are also preparing the model weights and will add them to the repository.
> > > - In the revised manuscript, we have also corrected the typos and incorporated the discussions you mentioned.
> > >
> > > These updates increase the impact of our dataset contributions and the MarsGen generator. By addressing your concerns, we trust our work is now more solid and hope these improvements are reflected in your final assessment.
> > >
> > > Sincerely,
> > >
> > > The Authors of Submission #317

---

### Note · Authors · 2025-08-15

Dear Chairs and Reviewers,

Thank you for your invaluable feedback. We appreciate your recognition of our paper's strengths and are glad our discussions resolved your concerns.

- **Reviewer xeWQ** praised our M3arsSynth for creating *\"physically accurate 3D Martian surface models\"* and MarsGen for producing *\"photorealistic and 3D-consistent video sequences of Martian terrains\"*.
- **Reviewer cDFy** highlighted the work's *\"Unique Dataset Theme\"*, *\"Diverse and High-Quality Data\"*, and *\"Robust Data Processing Pipeline\"*.
- **Reviewer Ehck** commended our method for creating the *\"first multimodal Mars dataset\"* by overcoming *\"textureless terrain and sparse viewpoints\"* and enabling *\"controllable video generation\"*, and recommended acceptance.
- **Reviewer cFSn** found our use of geometric foundation models a *\"promising and interesting direction\"* and the paper *\"well-written and easy to understand\"*.


The constructive discussion has strengthened our experiments and demonstrated that our work is more broadly applicable. We will incorporate the clarifications from the rebuttal stage into the revised version. We are confident in our final contribution and thank you for the insightful review process.

Sincerely,
The Authors

---

### Decision · Program_Chairs · 2025-09-18

**Decision:**

Accept (poster)

**Comment:**

This paper introduces Martian World Models, a novel framework for generating controllable Martian landscape videos. The system combines M3arsSynth, which processes NASA's sparse stereo images into high-fidelity 3D reconstructions (10K+ scenes), with MarsGen, a diffusion-based video generator that produces physically accurate and 3D-consistent Martian terrain sequences.

The research makes three significant advances: (1) creation of the first multimodal Mars dataset with synchronized 3D models, videos, and text descriptions; (2) development of a physics-accurate reconstruction pipeline using geometric foundation models; and (3) demonstration of controllable video generation from single images, camera trajectories, or text prompts.

During the discussion with the author and reviewers, the reviewers have reached a consensus on acceptance. And finally, we request that all promised revisions and exps should be added into the camera-ready version, especially for the ablation with MASt3R and Splatt3R, etc.
Reviewers appreciate the work's novelty and technical rigor, while noting limitations in environmental diversity and pipeline complexity. Authors addressed concerns through licensing documentation and methodological clarifications. The paper is accepted for its significant contribution, with suggestions to enhance dataset diversity and improve accessibility for future iterations.